# Study on the Allocation of a Rescue Base in the Arctic

**Yulong Shan**  **and Ren Zhang** *

College of Meteorology and Oceanography, National University of Defense Technology, Nanjing 210000, China
* Correspondence: Zrpaper@sohu.com; Tel.: +86-132-6078-6710

**Abstract:** Risk assessment and emergency responses to ensure the safety of ships crossing the Arctic have gained tremendous attention in recent years. However, asymmetry in the probability that people will receive aid when navigating through the Arctic still exists because of the unsystematic allocation of rescue bases in the Arctic. At the same time, no study has proposed an overall solution to the problem of allocating rescue bases in the Arctic region to safeguard people's interests. In this paper, we investigated the main natural factors affecting the safety of ship navigation in the Arctic based on the statistics of ship accidents in the Arctic from 1995 to 2004. The navigation risk of the Arctic was then assessed based on these natural factors, reflecting the need for rescue at all locations in the Arctic. Next, 37 cities with good infrastructure were selected among those along the Arctic as candidate locations for rescue bases. Finally, a new model was constructed based on the Set Covering Location Model, Double Covering Location Model, and P-Median Model to determine the optimal allocation of rescue bases in the Arctic. The rescue bases covered all the areas in the Arctic, and minimized cost in terms of distance and other economic factors. In addition, the constructed model ensured that two rescue bases were allocated to the areas with high navigation risk.

**Keywords:** Arctic; rescue base; Set-Double Covering Median Model

## 1. Introduction

The sea ice in the Arctic has been melting at rapid rates due to the rising levels of global warming [1]. According to the Fifth Assessment Report (AR5) of the Intergovernmental Panel on Climate Change (IPCC), it is highly likely that the Arctic sea ice will melt completely in late summer during the second half of the 21st century. Indeed, these conclusions were further confirmed by analyses of the Climate Model Intercomparison Project (CMIP) archives. This will lead to complete opening of the Arctic passages. However, the melting of sea ice is just one of the basic conditions that promote safe navigation in the Arctic. Given that the meteorological conditions of the Arctic are complex, there are many factors affecting the safety of ship navigation in the Arctic apart from sea ice. All these pose risks for objects navigating in the Arctic. Statistics indicate that more than 293 incidents occurred in the Arctic between 1995 and 2004 [2], which shows that emergency response and rescue strategies are required in the Arctic, given that people are navigating in the Arctic regions.

Recent years have witnessed an increase in the number of studies assessing the risks of Arctic navigation. Yet, few studies have focused on emergency response and rescue in the Arctic, especially the allocation of rescue bases in the Arctic. Presently, the number of rescue bases in the Arctic regions is not only small, but also each rescue base is built independently in the interests of the country building it, which not only leads to the waste of resources, but leaves many areas in the Arctic without a rescue bases, which in turn, causes asymmetry in the probability that people receive aid when navigating through the Arctic. To solve this problem, this study innovatively constructed a Set-Double Covering Median Model (SDCMM) to determine the best strategy for allocating rescue bases in the Arctic. We allocated rescue bases that not only covered all areas of the Arctic, but also minimized costs in terms of

distance and other economic considerations. In addition, the constructed model ensured that the two rescue bases covered the areas with high navigation risks.

Many scholars have proposed models for allocating rescue bases; however many of them have focused on allocating rescue bases inland with fewer in coastal areas [3]. In 1826, Von Thünen [4] wrote the book titled, "The Isolated State", which is regarded as the first study on the site-selection problem. The constructed model uses distance as a constraint to analyze the interaction between cities and land use, and explains the phenomenon of agricultural areas around cities. In 1909, Weber [5] addressed the industrial location problem. He investigated choosing the location of a warehouse so as to minimize the total distance between the warehouse and the customer. From then on, the problem of industrial layout was called the Weber problem. In 1964, Hakimi [6] wrote a paper on the P-Centre Model (PCM) for solving the challenge of minimizing the maximum distance between P facilities and all demand locations, which marked the beginning of modern theory on site selection. Drezner and Wesolowsky [7] proposed an algorithm to solve the PCM, which involved the numerical integration of ordinary differential equations and was computationally superior to methods using nonlinear programming. In 1965, Hakimi [8] proposed the P-Median Model (PMM) to minimize the product of the distance from P facilities to all demand points, and the demand of the demand points. Goldman [9] proposed an algorithm to solve the PMM with one server, which was based on a reduction procedure that may also yield useful simplification of problems involving general networks. Considering that solving the model with many servers needs too much computation, greedy algorithm is the most commonly used method to solve the model. Although the algorithm cannot guarantee that the obtained solution is the optimum solution, it can guarantee that the obtained solution is a satisfactory solution for policymakers [10].

Roth [11] and Toregas et al. [12] first proposed the Set Covering Location Model (SCLM) to choose the location of fire control centers and ambulances. This model mainly solves the problem of minimizing the number or cost of emergency service facilities when all demand points are covered. Roth [11] proposed one probabilistic approach to obtain optimal solutions for such large coverage problems, and his success supports the contention that this approach "may be of general applicability to various optimization problems". The heart of the algorithm is obtaining independent locally optimum solutions. Toregas et al. [12] used linear programming supplemented by the addition of a single cut constraint to solve the SCLM. Aly and White [13] developed the SCLM where the shortest time between the emergency facility point and the demand points is a random variable. The model mainly describes how to determine the address of the service facility while minimizing the number of service facilities needed, when the time from the demand point to its nearest service facility is ≤ to a specified value.

Considering budget constraints and other aspects, scholars further proposed the Maximum Covering Location Model (MCLM), which is a transformation of the SCLM. Church and ReVelle [14,15] pioneered the MCLM and explored strategies to maximize the covered demand points when the number of facilities and coverage radius is known. The MCLM is regarded as one of the most effective models for solving the site-selection problem. However, the key assumption of the model is that the coverage is binary, that is, any demand point is either completely covered or not covered. This assumption may result in some demand points not being covered. In the actual allocation of rescue bases, a rescue base should provide service for each demand point whether the demand point is within the specific coverage radius of the rescue base or not, otherwise greater losses may be incurred in areas intended. To address this problem, Berman and Krass [16] proposed the Generalized Maximum Covering Location Model (GMCLM), in which each demand point was covered, but the coverage degree was defined to be between [0–1].

In 1997, Ogryczak [17] proposed the bi-objective model (BOM) for selecting an emergency facility location, which takes the average distance and the maximum distance to the emergency facility into consideration, striking a compromise between the two indicators. In 1998, Badri et al. [18] established a multi-objective mathematical model (MOMM) to select the location of a fire station using a goal programming method. This model not only took the traditional time and distance into consideration,

but also considered the cost of building the fire station. In 2011, Canbolat and von Massow [19] studied the problem of emergency facility layout when the location of the demand point is random. The goal was to minimize the expected maximum straight-line distance from the facility to the demand point. In 2017, Li et al. [20] proposed the Double Covering Location Model (DCLM) to identify the location of a rescue base. Their model was based on the characteristics of traffic flow and the statistics of maritime traffic accidents in the Bohai Sea and used genetic algorithms to obtain the optimal locations of the rescue bases. The model takes time, distance, double covering, and the number of bases into account, where double covering means that each rescue base covers two demand points at most. The objective of the model is to minimize the construction and operation costs of the marine emergency rescue system, on the premise that it can cover all the demand points.

Based on the above description, many models have been proposed for selecting the best locations for setting up rescue bases. The main models and their respective characteristics are shown in Table 1. To choose the best location for a rescue base in the Arctic, this study used the PMM, SCLM, and DCLM to comprehensively construct the Set-Double Covering Median Model (SDCMM). The constructed model enables the determination of the number and location of rescue bases in the Arctic, under the conditions that all demand points are within the range of rescue base and the total distance of the rescue bases to all demand points and the total construction cost of all rescue bases are minimized. At the same time, the model can guarantee that two rescue bases cover the sea areas with high navigation risk.

**Table 1.** The main models for the optimal layout of rescue bases and their characteristics.

| Model | Objective Function | Deficiency | References |
|-------|-------------------|------------|------------|
| PCM | Minimizing the maximum distance from P rescue points to all demand points | Only considers the distance, and the number of rescue points needs to be determined in advance. | [6] |
| PMM | Minimizing the product of distance from P rescue points to demand points and the need of all demand points. | Only considers distance and demand, and the number of rescue points should be determined in advance. | [8] |
| SCLM | Minimizing the number or total cost of rescue points when all emergency points are covered | Does not take the needs of each demand point into consideration | [11,12] |
| MCLM | Maximizing the acceptable demand of the rescue bases on the premise that the number of facilities and coverage radius are known | Does not guarantee that all demand points can be covered | [14,15] |
| BOM | Minimizing the average distance and the maximum distance from the rescue bases to demand points. | Does not consider the needs of each demand point | [17] |
| DCLM | The construction and operation cost of the maritime rescue system is as low as possible when all demand points are covered. | Classifies the demand points into a few emergency points and not all demand points are covered. | [20] |

Overall, the objectives of this study were to (1) construct a new model to select the location of a rescue base, and (2) provide the optimal layout of rescue bases in the Arctic. The study is organized as follows: Section 2 introduces the technical process and modeling idea of this study. Section 3 describes the methods and results of the risk assessment for navigation in the Arctic. Section 4 presents the candidate rescue bases and estimation of their construction costs. The results are provided in Section 5 and the final section (Section 6) provides the conclusions and discussion.

## 2. Methods

### 2.1. Technical Process

To explore the main factors affecting navigation risk in the Arctic, this paper firstly describes the types and frequency of ship accidents in the Arctic, based on the limited statistical data available. Secondly, based on the main natural factors affecting navigation risk in the Arctic and the monthly average data of these factors in the past 17 years (2000–2016) in the Arctic, the navigation risk is calculated and the final risk values of each sea area are normalized. Next, the number and location of candidate rescue bases along the Arctic coast are determined, and the costs of building each of the bases into integrated rescue bases comprising airports, ports, and general hospitals are assessed on the basis of existing facilities. To reduce the cost of building a rescue base, the rescue bases selected in this paper currently have the necessary infrastructure such as airports, ports, and hospitals. Finally, based on the principles of PMM, SCLM, and DCLM, the number and location of rescue bases along the Arctic coast are determined. The selected bases are arrived at after considering the distance, construction costs, and the needs of each demand point, and they cover all areas in the Arctic. The flow chart of the technical process is shown in Figure 1.

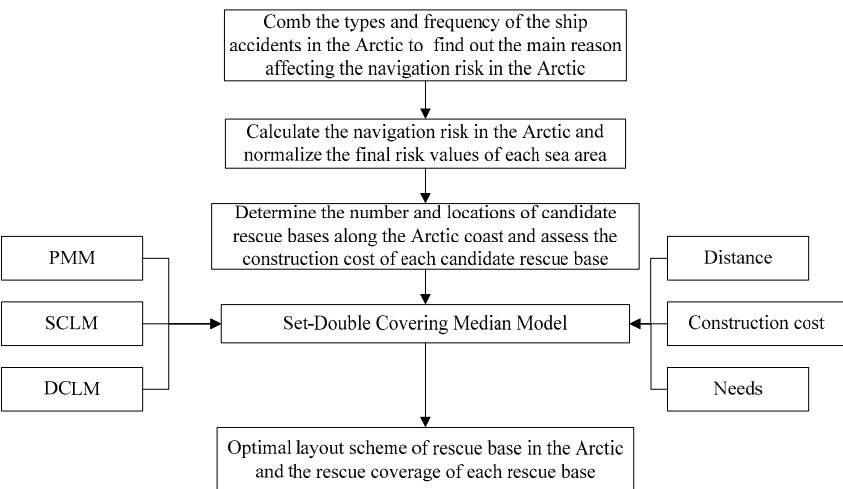

**Figure 1.** Flow chart of the technical process.

### 2.2. Set-Double Covering Median Model

The model established in this study is a combination of the PMM, SCLM, and DCLM, and is therefore named SDCMM. The core of the constructed model is the PMM. The following three conditions are required before running the PMM: the number and locations of demand points, the number and locations of candidate rescue bases, and the number of rescue bases to be built which is P [8]. Since this paper requires two rescue bases to be responsible for areas with high navigation risk, it is necessary to determine the locations that the two rescue bases are responsible for by calculating navigation risk values of each sea area in the Arctic when calculating the cost of each scheme using the evaluation function. The evaluation function of SDCMM is defined as:

$$min\left\{\sum_{i=1}^{N_1}\sum_{j=1}^{M} D_{ij} * y_{ij}^1 + \sum_{i=1}^{N_2}\sum_{j=1}^{M} D_{ij} * y_{ij}^2 + \sum_{j=1}^{M} C_j * x_j\right\}, \tag{1}$$

where $N_1$ is the number of demand points with low risk that only one rescue base is responsible for, $N_2$ is the number of demand points with high risk that two rescue bases are responsible for, $M$ is the number of candidate bases, $D_{ij}$ is the distance between the $i$th demand point and the $j$th rescue base, $C_j$ is the construction cost of the $j$th rescue base, $y_{ij}^1$ and $y_{ij}^2$ are used to determine whether the $j$th

rescue base is responsible for the *i*th demand point, and $x_j$ is used to determine whether to select the *j*th candidate base as the final rescue base.

$$y_{ij}^1 \text{ or } y_{ij}^2 = \begin{cases} 1, & \text{The } j\text{th rescue base is responsible for the } i\text{th demand point} \\ 0, & \text{The } j\text{th rescue base is responsible for the } i\text{th demand point} \end{cases}. \tag{2}$$

$$x_j = \begin{cases} 1, & \text{Select the } j\text{th candidate base as the final rescue base} \\ 0, & \text{Select the } j\text{th candidate base as the final rescue base} \end{cases}, \quad j \in [1, M]. \tag{3}$$

The constraints are defined as:

$$\sum_{j=1}^{M} y_{ij}^1 = 1, \tag{4}$$

$$\sum_{j=1}^{M} y_{ij}^2 = 2, \tag{5}$$

$$y_{ij}^1 \le x_j, \ i \in [1, N_1], \ j \in [1, M], \tag{6}$$

$$y_{ij}^2 \le x_j, \ i \in [1, N_2], \ j \in [1, M], \tag{7}$$

$$\sum_{j=1}^{M} x_j = P. \tag{8}$$

Formula (4) indicates the number of rescue bases allocated to a demand point with low risk that only needs one rescue base. Formula (5) indicates the number of rescue bases responsible for a demand point with high risk that needs two rescue bases. Formula (8) indicates that the number of the final selected rescue bases should be P. Formulas (6) and (7) indicate that if a candidate rescue base is not selected to be the final rescue base, the candidate rescue base will not be able to provide rescue services for the demand points. That is, if $x_j = 0$, then $y_{ij}^1$ or $y_{ij}^2 = 0$.

The model constructed above guarantees that, under the premise of planning to build P rescue bases, the selected rescue bases require minimal total costs including distance and economy, and the selected rescue bases can cover the whole Arctic sea area. To calculate the total cost of constructing base P while considering the distance and economic changes, we calculate the total cost under different P conditions and choose the best P with the lowest total cost as the final number of rescue bases to be constructed.

Since the dimensions of distance and construction costs are different, it is necessary to make them dimensionless. This is achieved using the method is shown in Formula (9). The final cost is calculated under the assumption that distance and economic cost have the same weight. Different weights for distance and economic cost are also considered to determine the priority objects when selecting the location of the rescue base, which depends on the wishes of decision-makers.

$$Data^1 = \frac{Data - Data_{min}}{Data_{max} - Data_{min}}, \tag{9}$$

where $Data^1$ is the dimensionless data, $Data$ is the raw data, $Data_{min}$ is the minimum value of the data set and $Data_{max}$ is the maximum value of the data set.

A greedy algorithm is useful to solve the model and the flow chart of the algorithm is shown in Figure 2.

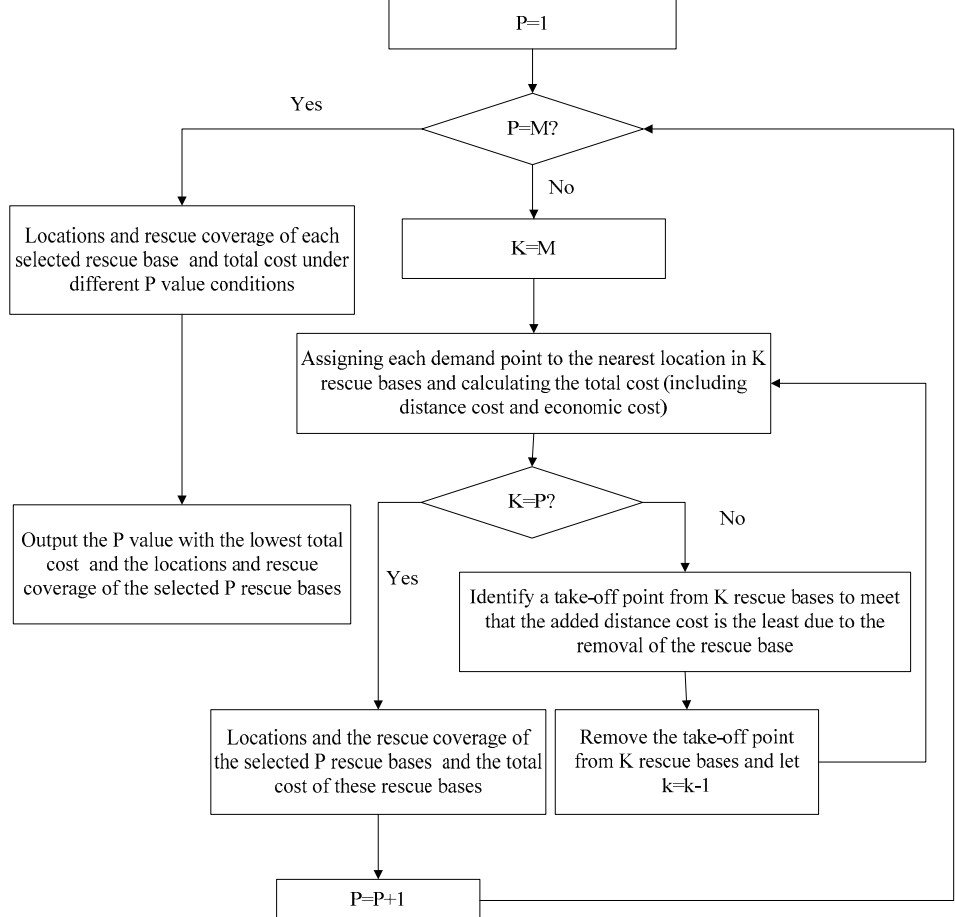

**Figure 2.** Flow chart of greedy algorithm to solve Set-Double Covering Median Model (SDCMM).

## 3. Navigation Risk in the Arctic

### 3.1. Statistics on Navigation Accidents in the Arctic

Since it is difficult to obtain the statistical data on navigation accidents in the Arctic region, this study relies on data collected between 1995 and 2004 contained in the Arctic Marine Shipping Assessment 2009 Report issued by the Arctic Council. The data come mainly from the Lloyds MIU (Marine Intelligence Unit) Sea Searcher database, the Canadian Hydraulics Centre Arctic Ice Regime System database, and the Canadian Transportation Safety Board. The details of the data are shown in Table 2.

**Table 2.** Statistics on the navigation accidents that occurred between 1995 and 2004 in the Arctic.

| Month | # | Vessel Type | # | Year | # | Primary Reason | # |
|-------|---|-------------|---|------|---|----------------|---|
| 1 | 16 | Bulk Carrier | 37 | 1995 | 35 | Collision | 22 |
| 2 | 35 | Container Ship | 8 | 1996 | 53 | Damage to Vessel | 54 |
| 3 | 30 | Fishing Vessel | 108 | 1997 | 23 | Fire/Explosion | 25 |
| 4 | 6 | General Cargo Ship | 72 | 1998 | 19 | Grounded | 68 |
| 5 | 15 | Government Vessel | 11 | 1999 | 21 | Machinery Damage/Failure | 71 |
| 6 | 18 | Oil/Gas Service & Supply | 1 | 2000 | 19 | Sunk/Submerged | 43 |
| 7 | 39 | Passenger Ship | 27 | 2001 | 31 | Miscellaneous | 10 |
| 8 | 22 | Pleasure Craft | 0 | 2002 | 30 | ——— | —— |
| 9 | 31 | Tanker Ship | 12 | 2003 | 28 | ——— | —— |
| 10 | 35 | Tug/Barge | 15 | 2004 | 34 | ——— | —— |
| 11 | 23 | Unknown | 2 | — | —— | ——— | —— |
| 12 | 23 | ——— | —— | — | —— | ——— | —— |

The data show that 293 ship accidents occurred in the Arctic during the 10-year period. Data analysis revealed that the most accidents occurred in July and the fewest in April. The most accident-prone vessels were fishing vessels and the least accident-prone vessels were yachts. The highest number of accidents occurred in 1996 and the lowest number occurred in the period from 1998 to 2000. The main causes of these accidents were machinery damage and machinery failure. Table 3 illustrates the main natural factors closely associated with the causes of the above accidents. The data reveal that the main factors affecting navigation risk in the Arctic are wind speed, atmospheric visibility, current speed, wave height, sea ice thickness, sea ice density, air temperature, and water depth.

**Table 3.** The main natural factors closely associated with the causes of the above accidents.

| Primary Reason | Natural Factors |
|----------------|-----------------|
| Collision | WS, Vis, CS, WH |
| Damage to Vessel | WS, SIT and SID |
| Fire/Explosion | AT |
| Grounded | Depth |
| Machinery Damage/Failure | AT |
| Sunk/Submerged | CS and WH |

Note that WS means wind speed, Vis means atmospheric visibility, CS means current speed, WH means wave height, SIT means sea ice thickness, SID means sea ice density, AT means air temperature and Depth means water depth in Table 3.

*3.2. Data Used to Assess the Navigation Risk in the Arctic*

The data used to assess the navigation risk in the Arctic were the monthly average data of $1° \times 1°$ from 2000 to 2016. The data for wind speed, sea ice density, and air temperature were from the ERA-Interim data sets retrieved from the European Centre for Medium-Range Weather Forecasts (ECMWF); current speed data were from the global ocean reanalysis data set ORA-S4 retrieved from the ECMWF; wave height data were from the ERA5 data sets retrieved from the ECMWF; sea ice thickness data were from the SODA 3.4.2 data sets; and water depth data were from the ETOPO1 data sets retrieved from the National Oceanic and Atmospheric Administration (NOAA). Given that it is not easy to retrieve historical gridded Vis data in the Arctic, this study used an artificial neural network (ANN) to generate the gridded Vis in the Arctic from 2000 to 2016, based on the results of Shan et al. [21]. The data used to train ANN were from the International Comprehensive Ocean-Atmosphere Data Set (ICOADS) retrieved from the National Climate Information Centre of the United States (NCDC).

3.2.1. ERA-Interim

The ERA-Interim data is the gridded data product distributed by the ECMWF which is obtained by reanalysis of observations and predicted products from the entire world. The ERA-Interim data is

a third-level generation product and its quality is significantly higher compared to the second-level generation product called ERA-40. Data cover the time period from 1979 to the present and are constantly updated. The data set has a variety of temporal and spatial resolution products, of which the minimum temporal resolution is 3 h and the maximum time resolution is 1 month. The minimum and maximum grid resolutions are $0.125° \times 0.125°$ and $3° \times 3°$, respectively.

### 3.2.2. ORA-S4

Leveraging the fully-drawn lessons from global atmospheric reanalysis technology, ECMWF organized the Global Ocean Reanalysis Program (ORA) and released a series of reanalyzed data products. ORA-S4 is the improved version of ORA-S3. ORA-S4 assimilates satellite altimeter observations with improved accuracy compared to prior datasets. Similarly, the ORA-S4 product contains a collection of five assimilation results. Its horizontal resolution is $1° \times 1°$ and it increases to 42 layers in a vertical direction. Its depth ranges from 5 to 5350 m. The corresponding vertical resolution dramatically changes from 10 m at the surface layer to about 300 m at the bottom layer. ORA-S4 covers the period from September 1957 to the present (updated every 10 days with a delay of 6 days). Currently, the time resolution of the downloadable products is presented as a monthly average, and includes five variables: salinity, temperature, latitudinal current velocity, meridional current velocity, and sea surface height.

### 3.2.3. ERA5

ERA5 is the latest generation of reanalyzed data created by Copernicus Climate Change Service (C3S), a body funded by the European Union and operated by ECMWF. The data set is the improved version of ERA-Interim, hence it contains more historical observation data, especially satellite data, and advanced data assimilation, as well as model systems. The variables provided by ERA5 will be increased to 240, including wave height, wave direction, and other variables provided by coupled wave models. Collectively, these data will facilitate more accurate analysis of past atmospheric and oceanic states. The spatial and temporal resolution of ERA5 are 31 km and 1 h, respectively.

### 3.2.4. SODA

Simple Ocean Data Assimilation (SODA) was first proposed by the University of Maryland in the 1990s. SODA is an early global ocean reanalysis data research program supported by the National Science Foundation (NSF), which is designed to provide a set of marine reanalysis products matching atmospheric reanalysis products for climate research. Given the continuous advancement and upgrading of the assimilation system, SODA has released several versions of its data sets. Presently, the SODA assimilation system has been updated to the third generation, among which SODA 3.4.2 is a widely used product. This data product includes more than ten variables, including temperature, salinity, density, and current velocity. It also contains three different time resolution data sets: 5-day average, 10-day average, and monthly average. The maximum spatial resolution of the data set is $0.5° \times 0.5°$.

### 3.2.5. ICOADS

ICOADS archives are the largest collection of ocean surface observational data sets covering the period from 1784 to the present, including data from ships, buoys, and coastal sites from all parts of the world and are distributed by the NCDC. Because of the nature of the sampling, its observational station density changes with time and location. Vis records come with a Vis level as described in ICOADS documents and by Gultepe et al. [22].

3.2.6. Visibility Data from ANN

This data set is generated on the basis of the research by Shan et al. [21]. In this data set, the relationship between visibility and its influencing factors is fitted by the BP neural network. The data used to train ANN are from ICOADS, while data used to generate gridded visibility are from ERA-Interim. The temporal resolution of the generated gridded visibility is one month, and its spatial resolution is $1° \times 1°$. Since the visibility data in ICOADS are recorded based on visibility level, the final visibility data generated in this study are the visibility level. The rules used to classify the levels of visibility are shown in Table 4.

**Table 4.** The rules for classifying the levels of visibility.

| Vis Level | 1 | 2 | 3 | 4 | 5 | 6 | 7 | 8 | 9 | 10 |
|---|---|---|---|---|---|---|---|---|---|---|
| Vis value (km) | ≤0.05 | 0.05~0.2 | 0.2~0.5 | 0.5~1 | 1~2 | 2~4 | 4~10 | 10~20 | 20~50 | ≥50 |

The download links of these data are shown in Table 5.

**Table 5.** The download links of the data used.

| Data | Data Set | Download Link |
|---|---|---|
| WS | ERA-interim | https://apps.ecmwf.int/datasets/data/interim-full-moda/levtype=sfc/ |
| CS | ORA-S4 | ftp://ftp-icdc.cen.uni-hamburg.de/EASYInit/ORA-S4/monthly_1x1/ |
| WH | ERA5 | https://cds.climate.copernicus.eu/#!/search?text=ERA5&type=dataset |
| SIT | SODA | https://www.atmos.umd.edu/~{}ocean/index_files/soda3.4.2_mn_download.htm |
| SID | ERA-interim | https://apps.ecmwf.int/datasets/data/interim-full-moda/levtype=sfc/ |
| AT | ERA-interim | https://apps.ecmwf.int/datasets/data/interim-full-moda/levtype=sfc/ |
| Depth | Etopo1 | http://maps.ngdc.noaa.gov/viewers/wcs-client/ |

*3.3. Navigation Risk in the Arctic*

Figure 3 shows the 17-year average of wind speed (WS), current speed (CS), wave height (WH), sea ice thickness (SIT), sea ice density (SID), air temperature (AT) and atmospheric visibility (Vis), as well as the depth of the Arctic (66° N–90° N). The data reveal that WS, SIT, SID, AT, Vis, and depth vary in different Arctic sea areas. Thus, we infer that these data can be used as a reference for navigation risk assessment in different Arctic sea areas. However, the difference in CS in different sea areas in the Arctic is small, being below 0.1 m/s. We suggest that its impact on navigation risk can be neglected. At the same time, it can be noted that the data for WH have many NaN values in the Arctic. Considering the proportional relationship between WH and WS, we did not incorporate WH when assessing the navigation risk in the Arctic. In conclusion, this study only takes WS, SIT, SID, AT, Vis, and Depth into consideration when assessing the navigation risk in the Arctic.

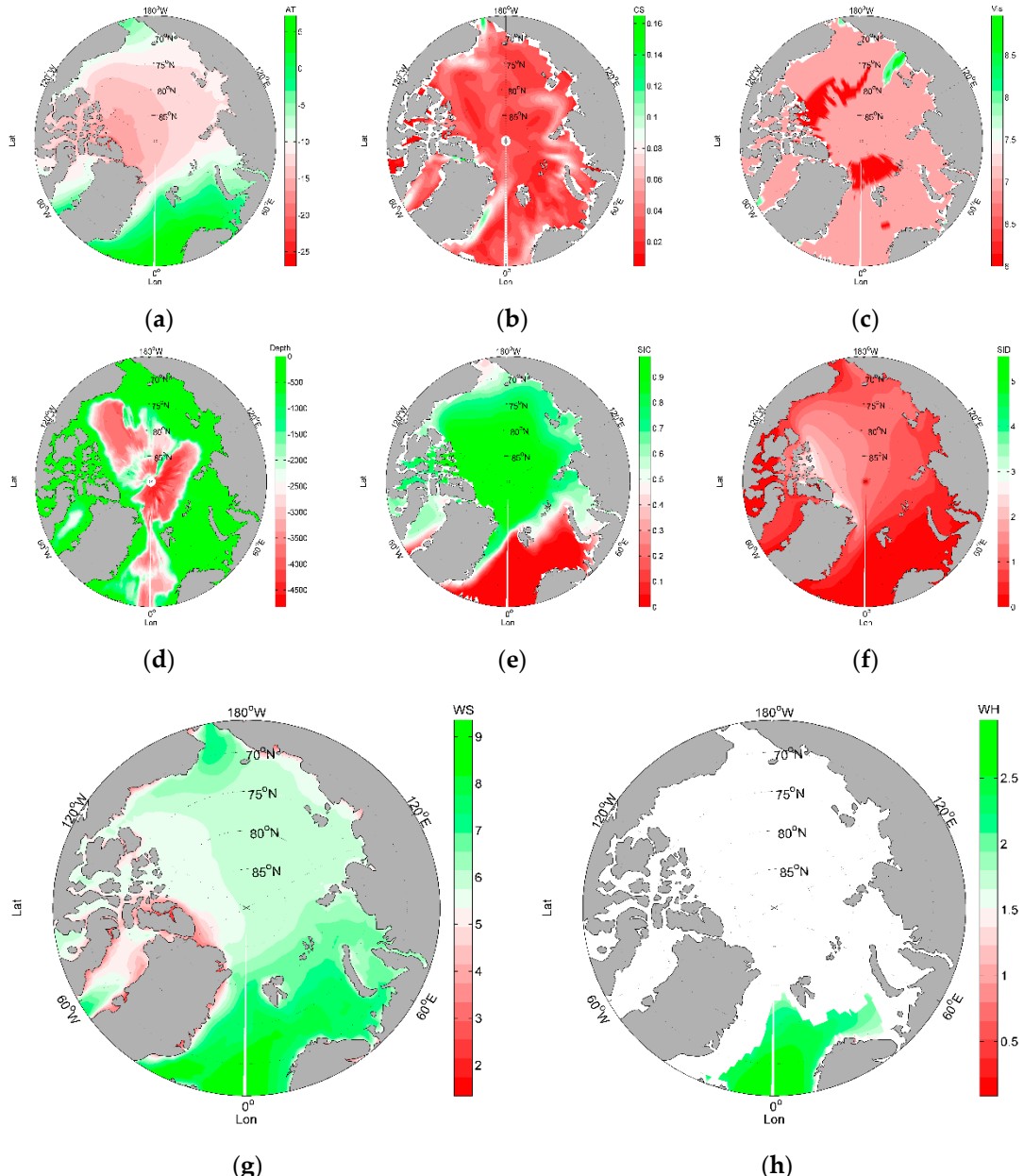

**Figure 3.** The 17-year average of (**a**) AT; (**b**) CS; (**c**) Vis; (**d**) the depth in the Arctic; (**e**) SIC; (**f**) SID; (**g**) WS and (**h**) WH.

It is important to eliminate the dimensions of the data before calculating the navigation risk in the Arctic. The method used to eliminate the dimensions of WS, SIT, and SID is shown in Formula (9); the method for eliminating the dimensions of Vis and AT is shown in Formula (10).

$$Data^1 = \frac{Data_{max} - Data}{Data_{max} - Data_{min}}. \tag{10}$$

In this study we assume that when the depth is greater than 50 m, the influence of depth on navigation safety can be neglected. The method used to eliminate the dimension of depth is shown in Formula (11).

$$\begin{cases} Depth^1 = 0 \ , \ Depth \geq 50 \\ Depth^1 = 1 - \frac{Depth}{50}, 0 < Depth < 50 \\ Depth^1 = 1 \ , \ Depth \leq 0 \end{cases}. \tag{11}$$

The method used to calculate the navigation risk in this study is derived from the research results reported by Li et al. [23]. The contribution weight of each factor to the navigation risk refers to the research results obtained by Yu et al. [24] (Table 6), which is a combination of the G1 method and the entropy method.

**Table 6.** The contribution weight of each factor to the navigation risk.

| Factors | WS | SIT | SID | AT | Vis | Depth |
|---------|------|------|------|------|------|-------|
| Weights | 0.1281 | 0.2734 | 0.1940 | 0.0682 | 0.1137 | 0.2226 |

The navigation risk value obtained is normalized using the method shown in Formula (10). Figure 4 shows the distribution of mean navigation risk in the Arctic from 2001 to 2016. It can be seen that the risk varies in different areas of the Arctic region. The navigation risks in the Kara Sea, the Barents Sea, the Norwegian Sea, the Greenland Sea, and the Bay of Baffin are lower than the risks in the East Siberian Sea. In this study, we assume that the locations with risks greater than 0.5 need two rescue bases.

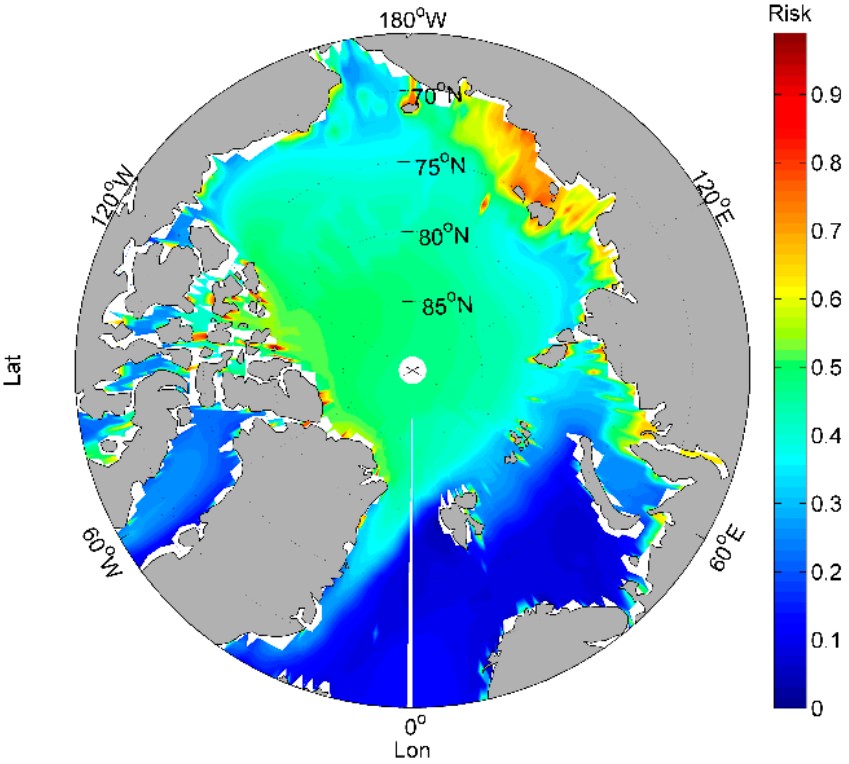

**Figure 4.** The distribution of mean navigation risk in the Arctic from 2001 to 2016.

## 4. Candidate Rescue Bases and Estimation of Their Construction Costs

### 4.1. Candidate Rescue Bases in the Arctic

The candidate rescue bases selected in this study were required to meet two conditions: (1) the locations of the candidate rescue bases should be north of 60° N and along the Arctic Ocean; (2) the candidate rescue bases should be cities or towns with airports, ports, and hospitals. If there were no cities or towns meeting the facility conditions in a large area, priority was given to the cities and towns with airports. A total of 37 cities and towns were selected to be the candidate rescue bases in the Arctic and the details of these candidate rescue bases are shown in Table 7. Figure 5 shows the distribution of these candidate locations. All the data shown in Table 7 were obtained from Wikipedia.

**Table 7.** The details of the candidate rescue bases in the Arctic.

| Order | Lon (°) | Lat (°) | City | Country | Description |
|---|---|---|---|---|---|
| 1 | −166.76 | 68.35 | Point Hope | United States (USA) | A city with good airport and port facilities and a health clinic in Alaska. |
| 2 | −165.4 | 64.5 | Nome | | A small port city with good airport, port, and highway facilities and a regional hospital and health center in Alaska. |
| 3 | −156.78 | 71.29 | Barrow | | The northernmost city in the United States, which has good airport and port facilities and big hospitals. |
| 4 | −148.71 | 70.33 | Prudhoe bay | | An oil field in the United States, which has good airport and port facilities and emergency medical services. |
| 5 | −133.03 | 69.44 | Tuktoyaktuk | Canada (CAN) | A fishing village in Canada with good airport, port, and highway facilities and a health center. |
| 6 | −125.25 | 71.99 | Sachs Harbour | | A village in the northwest of Canada with good airport and port facilities and a health center. |
| 7 | −95.88 | 68.63 | Gjoa Haven | | A village in Canada with good airport and port facilities and a medical and health care institution. |
| 8 | −94.83 | 74.7 | Resolute | | One of Canada's northernmost communities, with good airport and port facilities, and a health center. |
| 9 | −92.1 | 62.81 | Rankin Inlet | | A small village in Canada with good airport and port facilities and a medical center. |
| 10 | −86.24 | 66.52 | Naujaat | | A small village in Canada with airport and port facilities and a medical center. |
| 11 | −85.94 | 79.99 | Eureka | | One permanent Research Community with an airport. |
| 12 | −83.16 | 64.14 | Coral Harbour | | A community in Canada with good airport and port facilities and a health center. |
| 13 | −75.65 | 62.2 | Salluit | | A community in Canada with good airport and port facilities and a hospital. |
| 14 | −68.52 | 63.75 | Iqaluit | | A city in Canada with good airport, port, and highway facilities, a general hospital, and a family clinic. |
| 15 | −65.7 | 66.15 | Pangnirtung | | A small village in Canada with good port and airport facilities and a medical center. |
| 16 | −62.34 | 82.5 | Alert | | Permanent residence in the northernmost part of the world, with airport facilities. |
| 17 | −56.15 | 72.79 | Upernavik | Greenland (GL) | The thirteenth largest city in Greenland with good port and airport facilities and big hospitals. |
| 18 | −53.67 | 66.94 | Sisimiut | | The second largest city in Greenland with good airports, ports, and road facilities, and hospitals. |
| 19 | −52.87 | 68.71 | Aasiaat | | The fifth largest city in Greenland with good airports, ports, and road facilities, and a hospital. |
| 20 | −52.13 | 70.67 | Uummannaq | | The eleventh largest city in Greenland with good airports, ports, and road facilities, and a hospital. |
| 21 | −51.74 | 64.18 | Nuuk | | The capital of Greenland, with good airports, ports, highways, and a large hospital. |
| 22 | −23.13 | 66.08 | Isafjorzur | Iceland (ISL) | A small town in Northwest Iceland with good airports, ports, highways, and a hospital. |
| 23 | −21.93 | 64.13 | Reykjavik | | The capital of Iceland, with good airports, ports, highway facilities, and large hospitals. |
| 24 | −18.1 | 65.68 | Akureyri | | The fifth largest city in Iceland, with good airports, ports, and highway facilities, and a large hospital. |
| 25 | −15.21 | 64.25 | Hofn | | The second largest town in southeastern Iceland, with good airports, ports, highways, and two hospitals. |
| 26 | 14.38 | 67.28 | Bodo | Norway (NOR) | The capital and largest city of Norland, Norway. It has good airport and port facilities and many hospitals. |
| 27 | 18.94 | 69.68 | Tromssa | | The 18th largest city in Norway, with good airport and port facilities and a large hospital. |
| 28 | 23.68 | 70.66 | Hammerfest | | The 130th largest city in Norway, with good airport and port facilities and two hospitals. |

**Table 7.** *Cont.*

| Order | Lon (°) | Lat (°) | City | Country | Description |
|---|---|---|---|---|---|
| 29 | 33.08 | 68.97 | Murmansk | | The capital of the Molmansk in Russia, with good airports, ports, roads, railways, and many large hospitals. |
| 30 | 40.53 | 64.53 | Arkhangelsk | | The capital of Alhangersk in Russia, with good airports, ports, roads and railways, and a number of general hospitals. |
| 31 | 44.23 | 65.85 | Mezen | Russia (RUS) | A city in Russia with good airport and port facilities and a clinic. |
| 32 | 61.66 | 69.76 | Amderma | | A village in Russia with airport and port facilities. |
| 33 | 72.07 | 71.3 | Sabetta | | A village in Russia with good airports, ports, and railway facilities. |
| 34 | 80.52 | 73.5 | Dikson | | A city in Russia with good airport and port facilities and a hospital. |
| 35 | 128.87 | 71.65 | Tiksi | | A city in Russia with good airport and port facilities and a hospital. |
| 36 | 170.28 | 69.7 | Pevek | | The northernmost town in Russia, with good airports, ports, and highways, and a hospital. |
| 37 | −179.42 | 68.89 | Mys Shmidta | | A city in Russia with good airports, ports, and highways. |

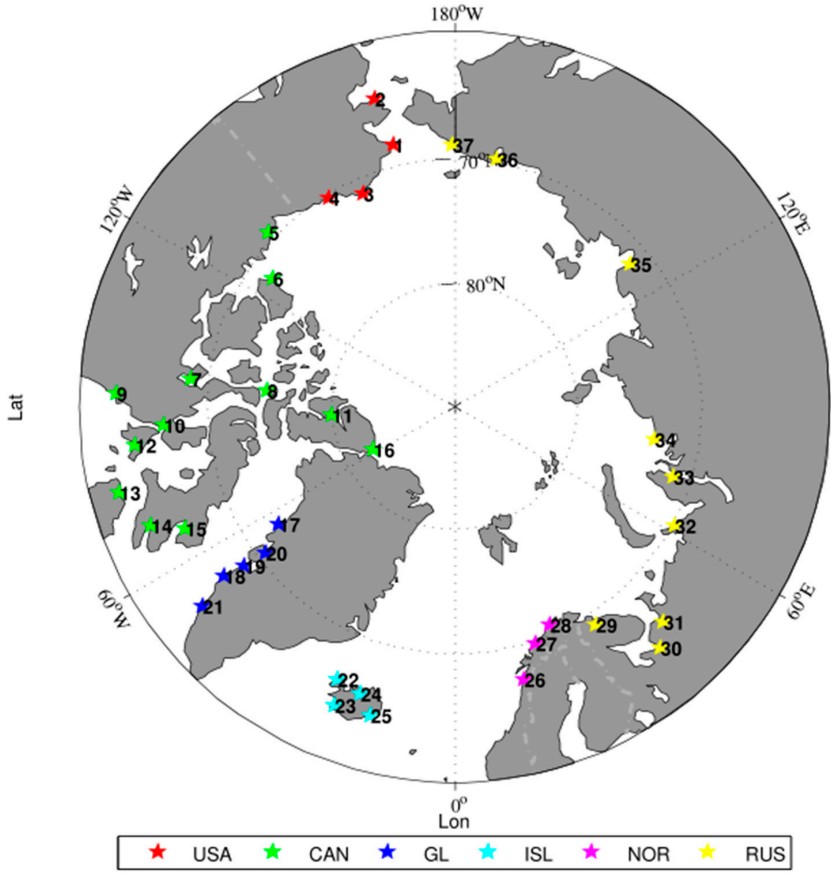

**Figure 5.** The distribution of these candidate locations.

## 4.2. Estimation of the Construction Cost of Each Candidate Rescue Base

Given the complexity of estimating the construction cost of each candidate rescue base, we utilized the level evaluation approach to evaluate the construction cost of each rescue base. The steps used were as follows:

Step 1: Assessment of airport facilities, port facilities, and medical conditions of each candidate rescue base was performed. For example, if the candidate rescue base did not have airport facilities, the level of the airport facilities for such a base was 1; if the base had airport facilities but the conditions of the airport facilities were not good, the level of the airport facilities for such a base was 2; if the

base had good airport facilities, then the level of the airport facilities for such a base was 3. A similar approach was used for the conditions of port facilities and hospitals in each candidate rescue base. The level was determined on the basis of the description of each candidate rescue base in Wikipedia.

Step 2: Assessment of the construction cost of each candidate rescue base. For example, if the level of airport facilities of the base was 1, then the construction cost of such facilities was 2. If the level of the airport facilities of the base was 2, then the construction cost of such facilities was 1. If the level of the airport facilities of the base was 3, then the construction cost of such facilities was 0. Similar assessments were performed for the construction cost of port facilities and hospitals in each candidate rescue base. The total construction cost of each candidate rescue base was the sum of its airport, port, and hospital construction costs.

Since the construction cost of each candidate base was also normalized in this study, the final result of the construction cost of each rescue base is reasonable. Table 8 shows the level of infrastructure and the total construction cost for each candidate base.

**Table 8.** The level of infrastructure and the total construction cost for each candidate base.

| Order | City | Level of Airport | Level of Port | Level of Hospital | Construction Cost | Normalized Construction Cost |
|---|---|---|---|---|---|---|
| 1 | Point Hope | 3 | 3 | 2 | 1 | 0.25 |
| 2 | Nome | 3 | 3 | 3 | 0 | 0 |
| 3 | Barrow | 3 | 3 | 3 | 0 | 0 |
| 4 | Prudhoe bay | 3 | 3 | 2 | 1 | 0.25 |
| 5 | Tuktoyaktuk | 3 | 3 | 2 | 1 | 0.25 |
| 6 | Sachs Harbour | 3 | 3 | 2 | 1 | 0.25 |
| 7 | Gjoa Haven | 3 | 3 | 2 | 1 | 0.25 |
| 8 | Resolute | 3 | 3 | 2 | 1 | 0.25 |
| 9 | Rankin Inlet | 3 | 3 | 2 | 1 | 0.25 |
| 10 | Naujaat | 3 | 3 | 2 | 1 | 0.25 |
| 11 | Eureka | 3 | 1 | 1 | 4 | 1 |
| 12 | Coral Harbour | 3 | 3 | 2 | 1 | 0.25 |
| 13 | Salluit | 3 | 3 | 3 | 0 | 0 |
| 14 | lqaluit | 3 | 3 | 3 | 0 | 0 |
| 15 | Pangnirtung | 3 | 3 | 2 | 1 | 0.25 |
| 16 | Alert | 3 | 1 | 1 | 4 | 1 |
| 17 | Upernavik | 3 | 3 | 3 | 0 | 0 |
| 18 | Sisimiut | 3 | 3 | 3 | 0 | 0 |
| 19 | Aasiaat | 3 | 3 | 3 | 0 | 0 |
| 20 | Uummannaq | 3 | 3 | 3 | 0 | 0 |
| 21 | Nuuk | 3 | 3 | 3 | 0 | 0 |
| 22 | Isafjorzur | 3 | 3 | 3 | 0 | 0 |
| 23 | Reykjavik | 3 | 3 | 3 | 0 | 0 |
| 24 | Akureyri | 3 | 3 | 3 | 0 | 0 |
| 25 | Hofn | 3 | 3 | 3 | 0 | 0 |
| 26 | Bodo | 3 | 3 | 3 | 0 | 0 |
| 27 | Tromssa | 3 | 3 | 3 | 0 | 0 |
| 28 | Hammerfest | 3 | 3 | 3 | 0 | 0 |
| 29 | Murmansk | 3 | 3 | 3 | 0 | 0 |
| 30 | Arkhangelsk | 3 | 3 | 3 | 0 | 0 |
| 31 | Mezen | 3 | 3 | 2 | 1 | 0.25 |
| 32 | Amderma | 3 | 3 | 1 | 2 | 0.5 |
| 33 | Sabetta | 3 | 3 | 1 | 2 | 0.5 |
| 34 | Dikson | 3 | 3 | 3 | 0 | 0 |
| 35 | Tiksi | 3 | 3 | 3 | 0 | 0 |
| 36 | Pevek | 3 | 3 | 3 | 0 | 0 |
| 37 | Mys Shmidta | 3 | 3 | 3 | 0 | 0 |

## 5. Results

### 5.1. Performance of Greedy Algorithm

Since the greedy algorithm is a local search algorithm and the result of each iteration is to find the local optimal solution, the final result of the greedy algorithm is likely to be the local optimal solution. At the same time, when using the greedy algorithm to solve the model constructed here, this paper assumed that all candidate bases were selected as the final result first, and then removed the candidate bases one by one, which minimized the added cost in each iteration until the number of remaining bases equaled the number of bases that needed be built (Figure 2). Therefore, the convergence speed of the greedy algorithm to solve the model depended on the number of bases that needed be built, which was the value of P. The time needed to obtain the optimal solution using the greedy algorithm with different P is shown in Figure 6. This shows that the calculation speed of the greedy algorithm is very fast, although it cannot guarantee that the obtained solution is the best solution. Note that the central processing unit of the computer used was i3-3227U (Intel Corporation, Santa Clara, CA, USA) and the random access memory capacity of the computer was 4G. The calculation is based on MATLAB 2104a software (MathWorks, Natick, MA, USA).

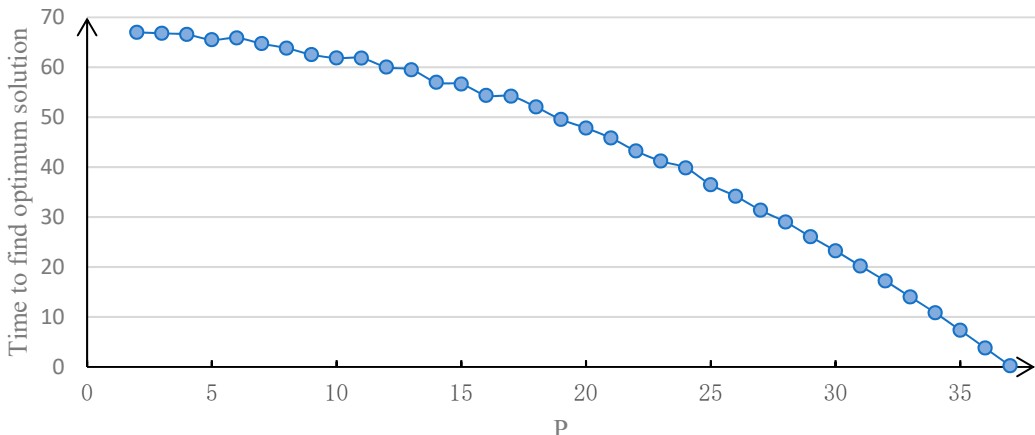

**Figure 6.** The time needed to obtain the optimal solution using greedy algorithm with different P.

### 5.2. Results of SDCMM

Figure 7 shows the total construction cost under different P. It can be seen that the least total construction cost appears when P = 35. At the same time, when P is less than 35, the total cost decreases with P. However, when P is more than 35, the total cost increases with P. The result is related to the value of the construction cost of each candidate rescue base and the weight of the distance cost and construction cost.

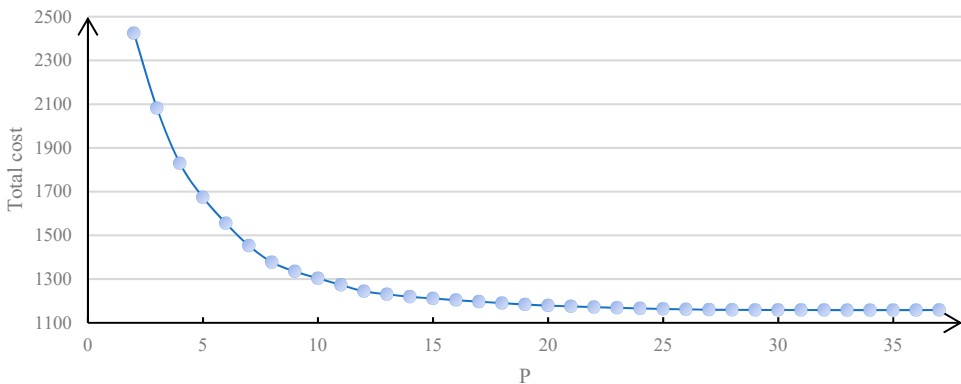

**Figure 7.** The total construction cost under different P.

Note that when P = 35, the selected rescue base "Reykjavik" is responsible for no need points, which is due to the construction cost for "Reykjavik" being 0. Since the approach used to calculate construction cost of each candidate rescue base in this paper is the simplified level evaluation method—which is not entirely reasonable because the construction cost of each candidate rescue base cannot be 0—the final number of selected rescue bases in this paper is 34, which does not include "Reykjavik". The locations of the selected rescue bases and their rescue coverage are shown in Figure 8. The serial number of each rescue base is shown in Table 8.

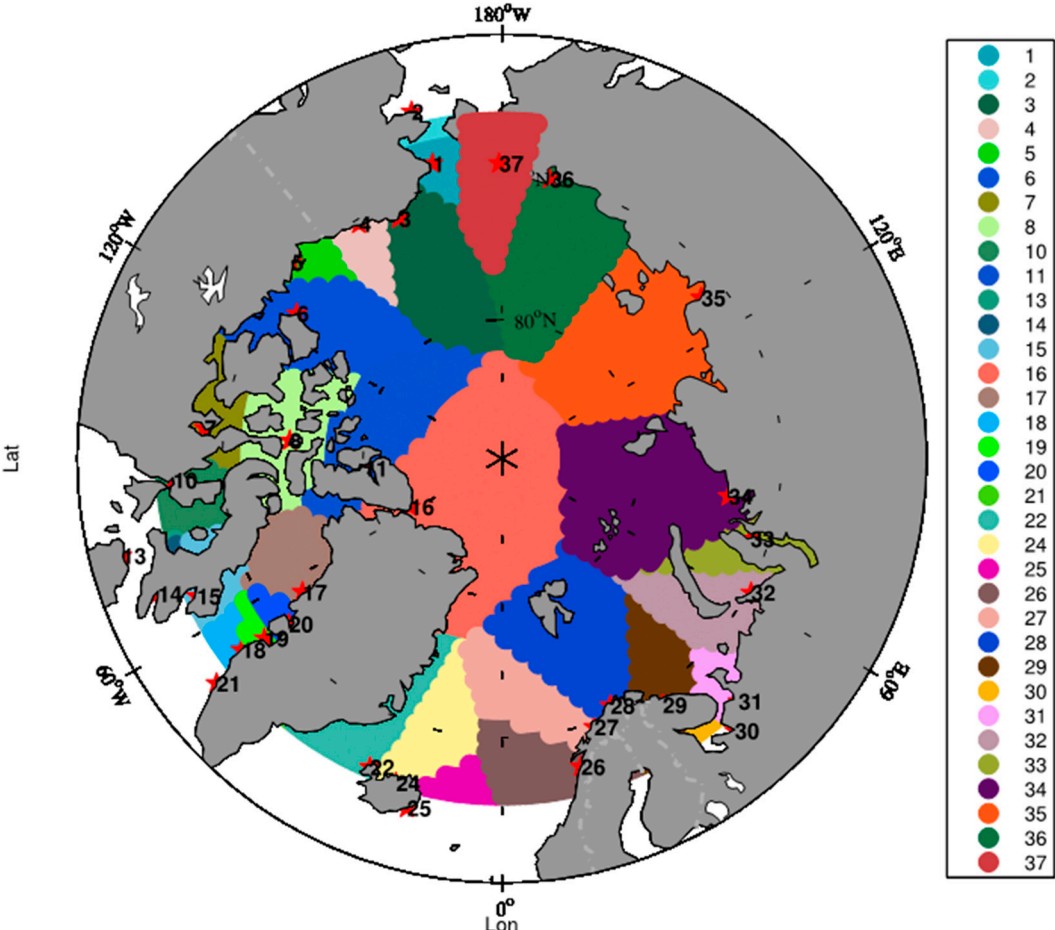

**Figure 8.** The locations of the selected rescue bases and their rescue coverage.

The need points that require two rescue bases to be responsible for them are shown in Figure 9 and details can be found in the Supplementary Materials.

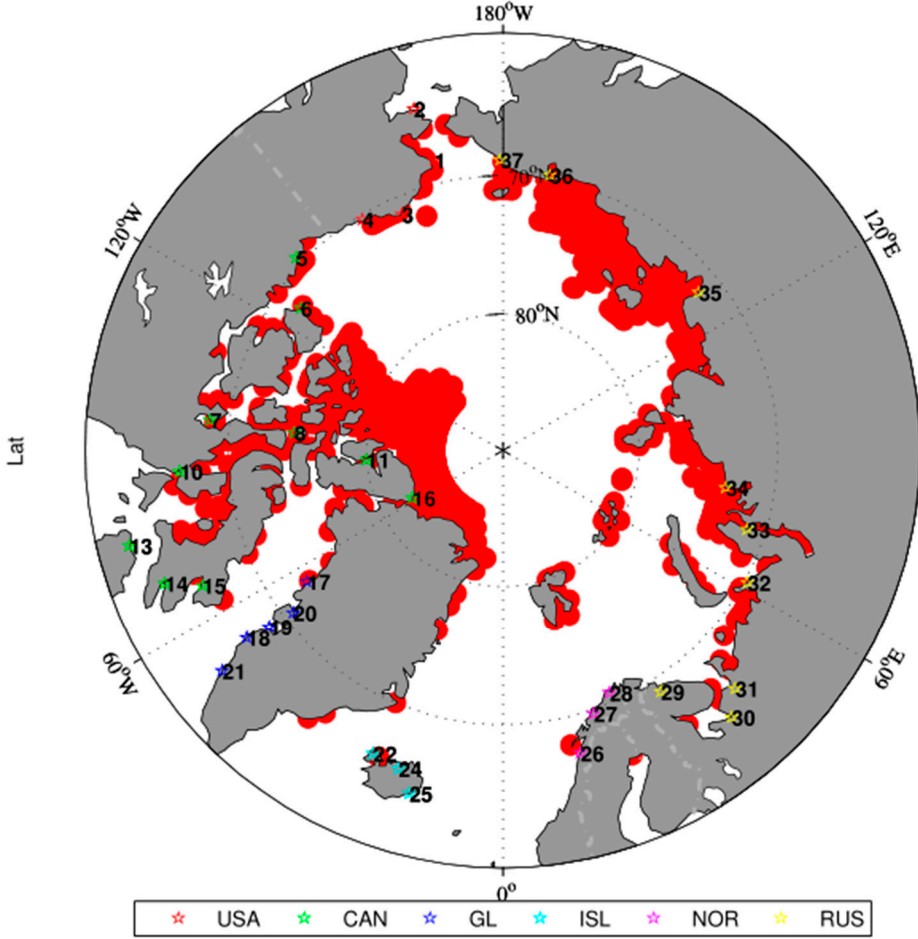

**Figure 9.** The need points that require two rescue bases to be responsible for them.

## 6. Conclusions and Discussion

As the Arctic passages gradually open, it is important to study the locations of Arctic rescue bases for countries around the Arctic region. In this study, a total of 37 candidate rescue bases were determined by consulting a large number of data sets. The SDCMM was innovatively constructed based on PMM, SCLM, and DCLM to determine the final locations of the rescue bases in the Arctic. The constructed model not only ensures that the selected rescue bases can cover all areas in the Arctic region, it also minimizes the total distance cost and the total construction cost of the rescue base from each demand point. In addition, two rescue bases are allocated to cover each demand point with a high navigation risk.

Although this study provides innovative methods to address the problem of allocating rescue bases in the Arctic, the following issues still exist:

(1) This study only considers the natural factors when assessing the navigation risk in the Arctic; however, this is incomplete as other non-natural factors may also affect the navigation risk. Moreover, the data used to calculate the risk value were the average of 17 years from 2001 to 2016, which may lead to neutralization for some areas with high risk. Therefore, it is important to assess the navigation risk by calculating the frequency of various extreme weather occurrences in each sea area to solve this problem. Additionally, statistical data on ship accidents should be integrated into the calculated risk data for each sea area. In this case, the risk value of the sea area where the accident occurred is expected to be high.

(2) The method used to estimate the construction cost of each candidate base in this study was relatively simple. Although the method reflects the current situation of each rescue base, the

difference in construction costs determined by this method is not significant. Hence, the estimated construction cost does not express the current situation of each candidate base sufficiently. Next, the construction standard and calculation of the construction cost of each rescue base should be discussed with relevant professionals in detail.

(3) The weights of economic cost and distance cost in the evaluation function of this study are the same. The next step is to explore the optimal layout of the rescue base under different weights for the two costs. The weights of the two costs will depend on the intention of the decision-maker.

(4) The algorithm to solve SDCMM is based on the idea of a greedy algorithm, which cannot guarantee that the obtained solution is the optimum solution. At the same time, although many new algorithms, such as genetic algorithms [20], can be used to solve conditioning optimization problems, it is also hard for them to obtain the optimum solution in a reasonable time in large-scale instances, and these algorithms also do not guarantee the accuracy of the solution theoretically [25]. With the development of artificial intelligence in recent years, deep learning has been used to solve many problems, such as conditioning optimization problems. Shoma et al. [25] applied deep learning and reinforcement learning to the "Traveling Salesman Problem" and obtained good results. Therefore, the next step is to apply deep learning and reinforcement learning to find the best solution with different P of the model constructed in this paper.

Overall, in this study, we innovatively constructed a SDCMM based on PMM, SCLM, and DCLM to determine the final locations of rescue bases in the Arctic. The study is of great significance in the context of global warming. Although there are some shortcomings in this study, we have discussed how to solve these limitations in the future. The authors plan to solve the shortcomings of this study and improve the allocation of rescue bases in the Arctic.

**Supplementary Materials:** The following are available online at http://www.mdpi.com/2073-8994/11/9/1073/s1, Table S1: the details of the need points that need 2 rescue bases be responsible for.

**Author Contributions:** Y.S. is the first author of this paper and the experiments and writing are mainly done by him. R.Z. is the corresponding author of the paper and he provided the ideas for writing the paper.

**Funding:** This research received no external funding and the APC was funded by National Natural Science Foundation of China (serial number item: 4197060570).

**Conflicts of Interest:** The authors declare no conflicts of interest.

## Abbreviations

| | |
|---|---|
| AT | air temperature |
| ANN | artificial neural network |
| AR5 | Fifth Assessment Report |
| BOM | bi-objective model |
| CS | current speed |
| C3S | Copernicus Climate Change Service |
| CMIP | Climate Model Intercomparison Project |
| Depth | water depth |
| DCLM | Double Covering Location Model |
| ECMWF | European Centre for Medium-Range Weather Forecasts |
| GMCLM | Generalized Maximum Covering Location Model |
| IPCC | Intergovernmental Panel on Climate Change |
| ICOADS | International Comprehensive Ocean-Atmosphere Data Set |
| MCLM | Maximum Covering Location Model |
| MOMM | Multi-Objective Mathematical Model |
| MIU | Marine Intelligence Unit |
| NOAA | National Oceanic and Atmospheric Administration |

| NCDC | National Climate Information Centre of the United States |
| NSF | National Science Foundation |
| ORA | Ocean Reanalysis Program |
| PMM | P-Median Model |
| PCM | P-Centre Model |
| SDCMM | Set-Double Covering Median Model |
| SCLM | Set Covering Location Model |
| SIT | sea ice thickness |
| SID | sea ice density |
| SODA | Simple Ocean Data Assimilation |
| Vis | atmospheric visibility |
| WS | wind speed |
| WH | wave height |

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
