# Peer review of "Study on the Allocation of a Rescue Base in the Arctic"

_symmetry, doi:10.3390/sym11091073_

Round 1
Reviewer 1 Report
The paper addresses an interesting theme in the context of the actual climate change around the world. I found the research question to be interesting. The literature review is quite well presented. As an observation, I believe that that the authors should argue more on the ideas they present in section 1 and 2, namely to add references for the statements they make, for example, in Table 1 please add a column with the references/ authors which have used these models in their papers. Also, in Table 3 please add just under the table the explanation of the acronyms (I saw that this was described just before the table, but I believe that it is easy to read it along with the table). Please explain what do you meant by: "The data used to assess the navigation risk in the Arctic are the monthly average data of 1°×1° from 2000 to 2016." in section 3.2. Please try not to use as many acronyms in the phrases, it is rather hard to read it - please try to explain in words or, where possible not to use the acronyms.
Reviewer 2 Report
The authors have analyzed the problem of optimal allocation of a rescue base in the Arctic region to safeguard the overall interests of humanity.
The authors propose a new model to select the location of a rescue base, by providing an optimal layout of rescue bases in the Arctic.
The works seems interesting.
The proposed SDCMM is based on previous PMM, SCLM and DCLM models referenced in the paper.
Anyway, the aauthors are suggested to recall some more details about previous mentioned models just to improve the overall comprehensibility of the mathematical algorithm SDCMM.
After that, the authors performed the navigation risk assessment.
Anyway, the proposed approach in the end it is a simple problem of constrained optimization than can be solved in different way.
The authors are suggested to include such indexes showing the performance and the convergence of the proposed approach. In few words, the authors are suggested to include such comparison with prior-art about the performance of the proposed approach in finding the optimal allocation of the rescue base: error metrics, std, convergence speed, etc...
Moreover, the authors did not analyze if recent Deep Learning approaches have addressed this issue.
For instance, the authors are suggested to investigate the use of such optimization algorithms based on recent Deep learning approaches for addressing the analyzed problem.
It is suggested to investigate if the usage of the following Deep Networks can improve the performance of the proposed approach with respect to the one proposed by authors:
Levenberg–Marquardt neural networks / Stacked AutoEconders
F. Rundo, S. Conoci, G. L. Banna, A. Ortis, F. Stanco and S. Battiato, Evaluation of Levenberg–Marquardt neural networks and stacked autoencoders clustering for skin lesion analysis, screening and follow-up, IET Computer Vision, vol. 12, no. 7, pp. 957-962, 10 201
Support Vector Machine
Anguita ; A. Boni ; S. Pace, Fast training of Support Vector Machines for regression, IEEE Proceedings International Joint Conference on Neural Networks. IJCNN 2000. Neural Computing: New Challenges and Perspectives for the New Millennium, 2000 Vol. 5;
Deep Learning and Reinforcement Learning
Shoma Miki ; Daisuke Yamamoto ; Hiroyuki Ebara, Applying Deep Learning and Reinforcement Learning to Traveling Salesman Problem, In IEEE Proceedings of the International Conference on Computing, Electronics & Communications Engineering (iCCECE) 2018, pp 65-70;
Although the above cited articles refer to different applications with respect to the problem analyzed by the authors, the optimization showed approaches can be applied to the problem suggested by the authors. Therefore, the authors are encouraged - by including a citation of the above reported papers - to test these approaches in solving the proposed problem or where it is not possible to do so, by arguing the use of these recent approaches in solving the proposed problem, identifying precise application ideas to be tested in a possible future works.
Round 2
Reviewer 2 Report
The authors did not full address the suggested revisions. It is reiterated to the authors to carefully follow the suggested revisions both in reference to the insertion of the details of the SCLM and DLCM methods and not only of the PMM and in reference to the explication of such algorithm performance indexes (even if referred to the PMM method) and to the discussion about the suggested papers (and not only) that describe the adoption of modern methodologies based on Deep Learning, which can be used to solve conditioning optimization problems such as those illustrated in the proposed article.
Round 3
Reviewer 2 Report
Ok, the paper is now ACCEPTABLE.